# Numerical Simulation and Validation of Laser Polishing of Alumina Ceramic Surface

**DOI:** 10.3390/mi14112012

**Published:** 2023-10-29

**Authors:** Chao Wang, Zhenyu Zhao, Houming Zhou, Junyong Zeng, Zhanwang Zhou

**Affiliations:** 1School of Sino-German Robotics, Shenzhen Institute of Information Technology, Shenzhen 518172, China; 202021002210@smail.xtu.edu.cn (C.W.); 202121542065@smail.xtu.edu.cn (Z.Z.); 2School of General Aviation, Jingchu University of Technology, Jingmen 448001, China; 3Mechanical Engineering and Mechanics, Xiangtan University, Xiangtan 411105, China; 4College of Mechatronics and Control Engineering, Shenzhen University, Shenzhen 518061, China

**Keywords:** laser polishing, alumina ceramics, orthogonal test, numerical simulation

## Abstract

Laser polishing is a noncontact and efficient processing method for surface treatment of different materials. It removes surface material and improves its quality by means of a laser beam that acts directly on the surface of the material. The material surface roughness is a major criterion that evaluates the polishing effect when alumina ceramics are polished by a laser. In this study, the effects of three factors, namely, laser power, scanning speed, and pulse frequency, on the surface roughness were investigated through orthogonal tests. The optimum polishing parameters were obtained through a comparison of the experimental results. Compared to the initial surface roughness (Ra = 1.624 μm), the roughness of the polished surface was reduced to Ra = 0.549 μm. A transient two-dimensional model was established by the COMSOL Multiphysics 5.5, and the flow condition of the material inside the molten pool of laser-polished alumina ceramics and the surface morphology of the smoothing process were investigated by utilizing the optimal polishing parameters obtained from the experiments. The simulation results showed that in the process of laser polishing, the fluid inside the molten pool flowed from the peaks to the valleys under the action of capillary force, and the inside of the molten pool tended to be smoothened gradually. In order to verify the correctness of the numerical model, the surface profile at the same position on the material surface was compared, and the results showed that the maximum error between the numerical simulation and the experimental results was 17.8%.

## 1. Introduction

Alumina is a structural ceramic that is widely applied in microelectronics, aerospace, and biomedicine, and it has the advantages of high hardness, high melting point, low density, and low thermal conductivity [1]. However, there are strict requirements of surface processing accuracy and surface finish for its applications. With the advancement of manufacturing technology, a novel noncontact laser polishing technique has emerged. It is considered to be a polishing technique with broad application prospects due to its high speed, excellent polishing quality, low environmental requirements, good adaptability to polishing objects, and little pollution to the environment [2]. It is expected to reduce the surface roughness of 3D-printed products with complex shape features by combining laser polishing with computer control and harnessing processing flexibility and low damage threshold [3,4].

Since the first use of CO_2_ lasers for polishing quartz surfaces in 1982 [5], many scholars have intensively investigated laser polishing processes for a variety of materials, including metals, glass, diamonds, and plastics [6,7]. Considering the high melting point and the low creep rate of ceramic materials, there are limited studies that focus on the laser polishing of ceramics. Bharatish et al. [8] used a 300 W CO_2_ pulsed laser for surface treatment of 92% alumina ceramic material and investigated the effect of polishing parameters on material removal rate and surface roughness. Folwaczny et al. [9] used four XeCI excimer lasers with different energy densities to irradiate four different dental ceramics. It was found that the laser energy density was inversely proportional to the surface roughness of the processed ceramics and that the laser energy density had a great impact on the quality of the polished surface. Nusser et al. [10,11] found that overlapping lasers could reduce or eliminate defects such as ripples and bumps on the material surface.

At present, it is difficult to observe a detailed physical evolution process as laser polishing often involves complex physical processes such as heat conduction, melting, evaporation, and cooling. To further study the flow condition in the melt pool during polishing and reduce the experimental cost and workload, some scholars have conducted simulation studies on the laser polishing process using numerical simulation. Perry et al. [12] established a transient one-dimensional heat conduction model to determine the minimum critical frequency by solving the maximum melting time and maximum melting depth. Vadali et al. [13] carried out a transient two-dimensional simulation that can predict the surface roughness after polishing and investigated the effect of laser pulse duration on fluid flow. Zhang et al. [14] established a two-dimensional numerical model to analyze the effects of thermocapillary force and capillary force on laser polishing in the melt pool by coupling heat transfer and fluid flow and simulated the flow process of the liquid material. Currently, in the field of laser polishing of ceramics, numerical simulations mainly focus on the analysis of temperature and stress fields. There is limited research concentrating on the evolution process of the melt pool for laser polishing of ceramics. Li et al. [15] investigated the effect of different heat sources on the size of the melt pool in the modeling of laser-melted ceramics. The simulation results showed that volumetric heat sources can predict the depth, width, and cross-sectional area of the melt pool more accurately than surface heat sources. Zhao et al. [16] simulated the temperature field and stress field of laser-ablated alumina ceramics using COMSOL and analyzed the thermal stress generation and crack expansion pathways.

In summary, scholars studying laser polishing of ceramics have mostly focused on experimental research, and numerical simulation of ceramic laser polishing is currently mainly focused on the temperature field and stress field analysis. There is less research on the evolution of laser-polished ceramic melt pool. The purpose of this study was to simulate the evolution of the material surface morphology under a moving heat source by combining experiments and numerical simulations using orthogonal tests to obtain the optimal polishing process parameters and establish a two-dimensional model. In order to verify the correctness of the numerical model, the surface profile at the same position after polishing was measured and compared with the surface profile obtained by simulation.

## 2. Orthogonal Tests

### 2.1. Material Properties

The experiment was carried out using 99 alumina ceramic (manufacturer: Shenzhen Hard Precision Ceramic Co., Ltd., Shenzhen, China) produced by conventional hot-press sintering. The specific chemical composition is shown in Table 1. This material is characterized by high hardness, high temperature resistance, high wear resistance, and lightweight. The material was cut into 146 mm × 95 mm × 5 mm ceramic plates using a hand-operated surface grinding machine before the experiment. Then, it was cleaned in an ultrasonic cleaner before polishing. The initial surface roughness was Ra = 1.624 μm.

### 2.2. Experimental Equipment

The equipment setup and schematic diagram of the laser polishing experiment are shown in Figure 1. The polishing equipment, shown in Figure 1a, consisted of a laser emitter, a dynamic focusing galvanometer, and a workbench. Figure 1b is the schematic diagram of the experiment. The laser used in the experiment was a pulsed CO_2_ laser (FSTI100SWC, SYNRAD, Seattle, DC, USA) with a power output range of 0–150 W, a pulse width of 150–300 μs, and an adjustable frequency range of 0–100 kHz. The laser galvanometer used was a dynamic focusing galvanometer (RF8330-3D-1200, Jinhaitron, Zhenjiang, China) with a focal length of 550 mm, a focused spot diameter of 0.314 mm, and a maximum scanning size of 400 mm × 400 mm. In this experiment, a white light interferometer (BRUKER WYKO Contour GT-K, Bruker, Billerica, MA, USA) and a metallurgical microscope (CX200E) were used to measure the surface roughness and observe the surface morphology. The white light interferometer had a resolution and accuracy of 0.1 nm, and the measurement area was 5 mm × 5 mm. Through comprehensive measures, such as environmental isolation, light source stability, optical path optimization, signal processing, calibration, appropriate operating techniques, and other measures, the uncertainty and noise of the white light interferometer measurements were effectively controlled to improve the accuracy and reliability of the measurement results.

### 2.3. Experimental Principle and Method

Before the polishing experiment, the initial roughness of the material surface Ra = 1.624 μm was measured using a white light interferometer. The prepared experimental sample was placed on a three-dimensional adjustment frame. After the focal length and the polishing parameters were set, the laser was turned on to polish the surface of the alumina ceramics. The polishing area was 10 × 10 mm^2^ each time. The polishing principle and polishing pathway are shown in Figure 2. To investigate the effect of processing parameters on the surface roughness, an L9 (3^3^) orthogonal table was established, as shown in Table 2, where *P* is the laser power, *v* is the laser scanning speed, and *f* is the laser pulse frequency.

### 2.4. Design and Result of the Orthogonal Test

The polishing experiment had three factors and three levels. A total of 27 sets of experiments were required for a full-scale experiment, and 9 sets of experiments were required when the experiment was arranged according to the orthogonal table L9 (3^3^). Therefore, the workload was significantly reduced. Table 3 shows the design matrix and result of the orthogonal test. The polished area of each test was 10 × 10 mm^2^. The surface roughness was measured three times after polishing, and the average value was calculated.

From Figure 3, it can be seen that the surface roughness reduction was more in Experiment 9, where the surface roughness was reduced by 63.125%, probably because the parameter combination of Experiment 9 was the closest to the optimal parameter combination, while Experiment 3 has the smallest reduction in surface roughness, with a surface roughness reduction of 46.25%.

### 2.5. Range Analysis for the Experiment

The experimental result was analyzed through range analysis to determine the optimal combination of factors. The fluctuation of factors in the corresponding column was observed through Range R. A larger R indicates that the factor had a greater effect on the surface roughness. Table 4 shows the range analysis for this experiment.

In Table 4, the R values are arranged according to average power *P*, scanning speed *v*, and pulse frequency *f*. A decline was observed in the effect of these factors on surface roughness. The average laser power and the pulse frequency directly determine the size of the energy density. The average power determines the size of a single laser pulse on the material surface, while the pulse frequency indicates the number of pulses per unit cycle. The energy density can be well selected only when the average power and pulse frequency are properly matched. The scanning speed *v* directly affects the duration of laser radiation on a material surface. The material surface will be subjected to short laser radiation if the scanning speed is too fast, causing the failure of material flow due to the unformed melt pool. The material surface will be subjected to excessive radiation due to an overly slow scanning speed. The material will be squeezed to spurt owing to back-flushing pressure caused by evaporation, resulting in a rougher surface [17]. Figure 4 describes the effect of the range of each factor on surface roughness.

With the minimum surface roughness value as the optimization objective, the best combination of levels was determined based on the results of the polar effect plot. The optimal combination, as observed from Figure 4, was average power *P* = 70 W, scanning speed *v* = 320 mm/s, and pulse frequency of laser *f* = 2 kHz. A supplementary experiment was conducted on this combination as it was not included in the aforementioned nine experiments. The surface roughness was Ra = 0.549 μm. Compared with the previous lowest surface roughness Ra = 0.59 μm, the surface roughness was further reduced, indicating that the polishing parameters were effectively optimized using orthogonal experiments. A white light interferometer was used to measure the material surface morphology in order to accurately reflect the changes that had occurred. The tested area was 5 × 5 mm^2^, as shown in Figure 5. Figure 5a shows the initial surface morphology of the material. The surface roughness was Ra = 1.624 μm. The height difference between the peak and the trough of the material surface was Rt = 61.99 μm. Figure 5b shows the three-dimensional morphology of the polished material surface. The surface roughness was Ra = 0.549 μm. The height difference between the peak and the trough of the material surface was Rt = 53.35. It was found that Ra decreased by 65.7%.

## 3. Numerical Simulation

### 3.1. Model Assumptions

To observe the flow condition of the material inside the molten pool of laser-polished alumina ceramics and the surface morphology smoothing process, a two-dimensional transient numerical model was established, and its specific modeling flow is shown in Figure 6 with the following assumptions on the model:(1)During laser polishing, the flow in the melt pool is considered as incompressible Newtonian laminar flow.(2)The distribution of material in the melt pool is characterized by isotropy and continuity. The parameters of material property are related to temperature only.(3)The purpose is to simplify the model and ignore the effect of evaporation and plasma on the melt pool flow.(4)The effect of slight laser defocusing on the surface is ignored owing to even material surface. The heat is input as a boundary heat source.

### 3.2. Material Property

The material used for the simulation was 99Al_2_O_3_ ceramic. During laser polishing, the phase change occurs on the material surface due to continuous laser radiation. The material can absorb or unleash heat upon the occurrence of phase change, which further impacts the temperature of the material surface, the size of the melt pool, and the fluid flow rate. Thus, temperature-dependent properties of alumina ceramics must be taken into consideration [18]. Figure 7 shows the variation of thermal conductivity and density with temperature. Table 5 presents other material property parameters and laser polishing parameters.

**Figure 7 micromachines-14-02012-f007:**
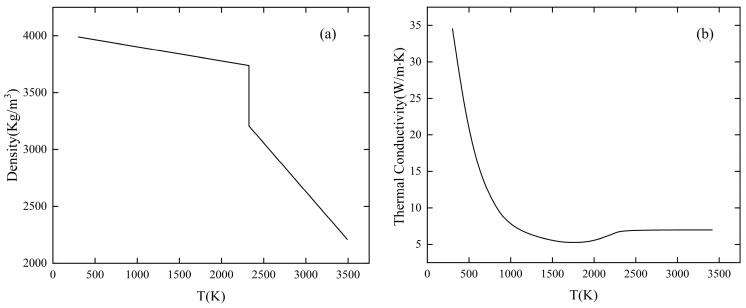
Variation of thermal conductivity and density with temperature [19]. (**a**) Density, (**b**) thermal conductivity.

**Table 5 micromachines-14-02012-t005:** Characteristics of alumina ceramics and parameters of laser polishing.

Parameter	Nomenclature	Value	References
Solidus temperature (K)	*T_s_*	2312	[20,21,22]
Liquidus temperature (K)	*T_l_*	2340	[20,21,22]
Melting temperature (K)	*T_m_*	2324	[20,21,22]
Evaporation temperature (K)	*T_v_*	3273.15	[20,21,22]
Ambient temperature (K)	*T_a_*	298.15	-
Dynamic viscosity (Pa s)	*μ*	10^5^ (*T* < *T_s_*) and 0.03 (*T* > *T_m_*)	[23]
Solidus specific heat (Jkg^−1^ K^−1^)	*C_p-s_*	1250	[19]
Liquidus specific heat (Jkg^−1^ K^−1^)	*C_p-l_*	1280	[19]
Convective coefficient (Wm^−2^ K^−1^)	*h*	100	[24]
Temperature derivative of surface tension (Nm^−1^ K^−1^)	*∂γ/∂T*	−8.2 × 10^−5^	[23]
Latent heat of melting (Jkg^−1^)	*L_m_*	1.067 × 10^6^	[20,21,22]
Emissivity	*ε*	0.7	[25,26]
Absorptivity	*α*	0.75	[27]
Spot radius (mm)	*r* _0_	0.157	-
Pulse width (μs)	*P_w_*	200	-
pulse repetition rate (kHz)	*f*	2	-
Laser power (W)	*P*	70	-
Laser scanning speed (mm/s)	*v*	320	-

### 3.3. Governing Equation

The model was simulated based on COMSOL. The temperature equation under the transient temperature field is expressed as follows [28]:(1)$$\rho C_{p}^{*}[\frac{\partial T}{\partial t}+\nabla \cdot (\vec{u}T)]+\nabla \cdot q=0$$
(2)q=−k∇T
where ρ is the density of alumina ceramics; T is the surface temperature during polishing; t is the polishing time; k is the thermal conductivity; $\vec{u}$ is the velocity vector, which is the variable coupled to the NS equation; and ∇ is the Hamiltonian operator. Equation (2) is the defined heat flux vector q using Fourier’s law of thermal conductivity. During polishing, the phase change occurs on the material under the action of laser radiation. The latent heat should be considered during phase change. Thus, the equivalent specific heat capacity $C_{p}^{*}$ was introduced herein:(3)$$C_{p}^{*}=C_{p-s}+L_{m}(\frac{df_{L}}{dt})$$
where $C_{p-s}$ is the specific heat capacity of solid alumina, $L_{m}$ is the latent heat of fusion for the transformation of alumina from the solid phase into the liquid phase, and $f_{L}$ is the volume fraction of the liquid phase in the melt pool and expressed as follows:(4)$$f_{L}=\left\{\begin{array}{c} \begin{array}{cc} 0 & T\leq T_{s} \end{array}\\ \begin{array}{cc} \frac{T-T_{s}}{T_{l}-T_{s}} & T_{s}\leq T\leq T_{l} \end{array}\\ \begin{array}{cc} 1 & T\geq T_{l} \end{array} \end{array}\right.$$
where $T_{s}$ and $T_{l}$ are the solidus temperature and liquidus temperature of alumina, respectively.

In order to solve for the fluid flow velocity, a nonisothermal flow interface was used to couple the heat transfer field and the flow field in the model. The fluid was assumed to be incompressible. Hence, the Navier–Stokes equation (NS) was derived as follows [18,19,20,21,22,23,24,25,26,27,28]:(5)$$\rho \left(\frac{\partial \vec{u}}{\partial t}+\left(\vec{u}-{\vec{u}}_{m}\right)\cdot \left(\nabla \vec{u}\right)\right)=\nabla \cdot \left(-pI+\mu \left(\nabla \vec{u}+{\left(\nabla \vec{u}\right)}^{T}\right)\right)+F_{V}$$
(6)$$F_{V}=\rho_{ref}g\left(1-\beta \left(T-T_{ref}\right)\right)$$
where p is the internal fluid pressure, I is the unit matrix, ${\vec{u}}_{m}$ is the kinematic velocity of mesh, and μ is the dynamic viscosity. $T_{ref}$ is the reference temperature, $\rho_{ref}$ is the reference density, β is the coefficient of thermal expansion, and g is the gravitational constant.

The NS equation was solved with the continuity equation.
(7)$$\nabla \cdot \vec{u}=0$$

The NS equation represents the conservation of momentum, while the continuity equation represents the conservation of mass.

In order to simulate the evolution of the surface morphology, a dynamic mesh was used to describe the internal fluid flow. Meanwhile, Laplacian smoothing was used in the calculation to improve the convergence of the numerical model. The governing equation of the mesh is expressed as follows:(8)$$u_{mesh}\cdot \vec{n}=u_{mat}\cdot \vec{n}$$
where $u_{mesh}$ is the dynamic mesh velocity, and $u_{mat}$ is the material velocity calculated from the NS equation.

### 3.4. Boundary Conditions

The white light interferometer was used to measure the initial surface profile in order to make it matching in both the simulation and experiment. Singularities will appear in the calculation if the initial surface profile is directly imported into the model as the collected initial profile has high-frequency noise. Therefore, the surface profile was Fourier processed before being imported into the model. Figure 8 shows the surface profile before and after processing. The red and black colors refer to the unprocessed and processed surface profile, respectively. The processed surface profile was imported into the model. A two-dimensional model was established, as shown in Figure 9, considering that the material only melts on the shallow surface during laser polishing. The model was 1000 μm × 300 μm in length and width. The boundary conditions in the model are shown in Table 6, where Boundary No. 1, 4 is surface-to-ambient radiation, and the boundary heat source is loaded on Boundary No. 1 with the same flat-topped beam as in the experiment.

### 3.5. Mesh Delineation

The model adopted a nonisothermal flow coupled with temperature field and flow field based on the governing equation and boundary conditions. The material melts and flows only on the shallow surface during laser polishing. Thus, a general physical calibration was used for the upper surface to allow finer mesh delineation. A hydrodynamic calibration was used for the rest of the area. In order to improve the computational speed, the mesh delineation was less fine than that of the upper surface. Automatic delineation was applied to prevent excessive mesh distortion during calculation. Table 7 shows the parameters of the mesh size in this model. The meshing results are shown in Figure 10.

## 4. Analysis of the Simulation Result

The model was used to simulate the evolution of laser polishing with a pulse width of 200 μs and a laser power of 70 W. The parameters of laser polishing were optimized and obtained from the aforementioned range analysis. Figure 11 shows the loaded square wave function before the laser heat source, simulating the effect of pulsed laser heating. The square-wave function had a heating time of 0.2 ms and a cooling time of 0.3 ms.

Figure 12a,b shows the temperature of the material surface at pulse termination time and starting time. Each time when the laser pulse is terminated, there exists a maximum surface temperature on the radiated area. The laser radiation will stop and the surface temperature will decline when the pulse termination time is exceeded. When the temperature decreases to the next pulse starting time, it will drop to the minimum value. As shown in Figure 12a, when the first laser radiated on the material, i.e., t = 0.2 ms, the material surface heated up rapidly and reached a temperature of 2200 K. This was followed by a decreasing surface temperature on the radiated area due to the terminated pulse. When t = 0.5 ms, the surface temperature dropped to 1050 K. When the second laser radiated on the material, i.e., t = 0.7 ms, the maximum temperature reached 3100 K because of the pulse heat. When the maximum temperature of the material surface exceeds the material melting temperature, the melt pool will initially take shape in the radiation area. When the third laser radiated on the material, i.e., t = 1.2 ms, the surface temperature did not rise significantly. Due to the moving heat source in the model, a dynamic equilibrium was reached between the laser heat input and laser movement speed, and the temperature of the material surface remained stable under laser radiation. Then, the temperature slightly decreased under the pulse action owing to the presence of surface-to-ambient radiation and heat convection. The bump observed on the temperature curve was attributed to the temperature fluctuation caused by prepulse. Furthermore, the temperature at the pulse termination time was less than the material evaporation temperature, which conformed to the assumption of Model (3). Only a small part of the material surface temperature was close to the melting temperature at the pulse starting time, as shown in Figure 12b, because the surface-to-ambient radiation and heat convection caused the surface temperature to decrease significantly after the laser radiated on the material surface.

During laser polishing, the curvature of the material surface profile was analyzed to reflect the polishing effect. As shown in Figure 13, a greater surface curvature indicates a larger fluctuation of material surface profile and rougher material surface. As the laser spot moves from left to right, the surface contour of the area not irradiated by the laser has the same spatial curvature at different moments, so it will cause some of the spatial curvature curves to overlap. The surface curvature of the melt pool was lower than the initial surface curvature at 1, 2, and 3 ms of laser polishing. This indicates that the material surface was smoothened by laser polishing. It can be noted that 0 and 4 ms indicate the spatial curvature of the surface profile before and after polishing, respectively. Comparatively, the curvature of the surface profile and the surface undulation decreased after polishing. It has been found that the surface profile curvature changes abruptly due to a noncontinuous pulsed laser. There will be small ripples after the melt pool cools down.

Figure 14 shows the evolution of the surface profile during laser polishing with a moving pulse heat source. Figure 14a–g shows the material surface profile and the fluid flow in the melt pool at 0.7, 1.2, 1.7, 2.2, 2.7, 3.2, and 3.7 ms, respectively. The white curve is the isotherm of the melting temperature. The authors of [28] showed that during laser polishing, the fluid in the melt pool is mainly driven by capillary force and thermocapillary force. The capillary force (surface tension) is positively related to the spatial curvature of the material surface in the direction normal to the free surface. The thermocapillary force is related to the temperature gradient on the material surface in the direction tangential to the free surface, with the temperature gradually decreasing from the center to the edge of the melt pool and the surface tension increasing with the temperature. Due to the negative surface tension temperature coefficient of alumina ceramics, the fluid in the melt pool flows from the temperature center to the edge [29]. In general, the capillary force is greater than the thermocapillary force in the early period of laser polishing owing to the large curvature of the surface profile and the small temperature gradient. Thus, the fluid is mainly driven by capillary force. When the raised surface profile is basically smoothed, the intensity of the capillary force is significantly reduced, while the internal temperature of the melt pool is significantly increased and the temperature gradient increases. At this time, the fluid flow is mainly driven by thermocapillary force [30].

The material surface was pulsed twice at t = 0.7 ms (Figure 14a). The material surface temperature reached up to 3100 K. The melt pool was 180 μm in width and 11 μm in maximum depth. Considering that the light spot was moving from Boundary 2 to the right side, the whole light spot was not completely radiated on the material surface. Hence, the width of the melt pool was much smaller than the spot diameter. Meanwhile, it was observed that the depth of the right side of the melt pool was much smaller than that of the left side. The temperature center in the melt pool was close to the left side because the front melt pool was radiated for a shorter period of time. The surface material flowed from the area with large undulation to the area with small undulation under capillary force and then rose under buoyancy. As shown in Figure 15a, the melt pool was mainly dominated by thermocapillary force at the left side. However, the rest of the area in the pool was dominated by capillary force resulting from the surface curvature. Most of the material surface was gradually smoothened by the capillary forces. The maximum flow velocity was 0.15 m/s in the melt pool. The laser spot was completely radiated on the material surface at 1.2 ms. Figure 14b shows that the smooth surface was realized in the melt pool by capillary force. It should be noted that backflow occurs in the tail of the melt pool. This is because the material in the melt pool flows to the left side, then to the solid–liquid boundary, and then slides down under gravity. The maximum flow velocity in the melt pool was 0.09 m/s. The right side of the melt pool exhibited a more undulating profile than the left side at 1.7 ms (Figure 14c). The fluid flowed from the raised area to the sunken area with velocity rising to 0.13 m/s. The light spot ran into the area with less undulation at 2.2 ms (Figure 14d). The maximum fluid velocity dropped to 0.12 m/s compared to 1.7 ms. The light spot again ran into the area with greater undulation at 2.7 ms (Figure 14e), with the maximum fluid velocity increasing dramatically to 0.23 m/s. At 3.2 ms (Figure 14f), the raised area on the surface profile was mostly smoothed, with the maximum fluid velocity down to 0.15 m/s. At 3.7 ms (Figure 14g), the right side of the light spot exceeded Boundary 4, with the width of the melt pool decreasing. The fluid flowed from the right side with a higher profile to the left side, with a maximum fluid flow velocity of 0.1 m/s. By comparing the flow velocity with time in Figure 14, it can be concluded that the maximum fluid flow velocity is related to the surface profile undulation. When the surface profile undulates, the maximum fluid flow velocity occurs in the melt pool. With greater profile undulation, the spatial curvature of the surface in that area increases, thus leading to a larger capillary force driving the fluid flow.

Figure 15 reflects the distribution of capillary forces and thermocapillary forces in the melt pool at different times. As the laser spot continued to run, capillary force was caused by the surface curvature and thermocapillary force was caused by temperature gradients. Considering the pulsed laser loaded in this model, the material surface cooled down as the laser did not radiate on the surface continuously. Compared to the continuous laser, the pulsed laser had a smaller heat accumulation. Moreover, the alumina ceramic material is much less thermally conductive than the metal. Thus, the melt pool was shallow after laser polishing. The Marangoni convection was rarely observed.

Figure 15a–g shows that at various moments, the large part of the area inside the molten pool was dominated by capillary forces. From this, it can be concluded that the fluid in the molten pool in the polishing stage was mainly driven by capillary forces, and the surface profile was gradually smoothened by the capillary forces as the dominant force, which consequently led to a gradual reduction of the surface roughness. However, the thermocapillary force was greater than the capillary force in a small proportion because, on the one hand, the capillary force significantly decreased with reducing curvature and, on the other hand, the thermocapillary force prominently increased due to the large temperature gradient in the corresponding area.

## 5. Verification of Experiment

Because of the assumed simplification of the process in the model, it was necessary to verify the simulation results through processing experiments. The processing experiments used the same laser process parameters as the numerical simulation and used a white light interferometer to test the surface morphology. Figure 16 shows the change in the contour curve at the same location through numerical simulation and experimental polishing. At the same location, a change in the contour curve was seen after laser polishing, and the surface bulge position was gradually smoothened. About 350 μm from the starting point of the position of the numerical simulation and experimental polishing, the depression maximum depth was −7.9 and −6.8 μm, respectively. About 420 μm from the starting point of the position of the numerical simulation and experimental polishing surface contour, the maximum height of the convexity was 4.67 and 5.5 μm, respectively. The maximum depth of the depression and the convexity of the maximum height had errors of 13.9% and 17.8%, respectively. Considering that the model was simplified with certain assumptions, the trends of surface profile changes obtained from the numerical simulation and experiments were roughly the same. In summary, the surface profiles of laser-polished alumina ceramics using numerical simulation and experiments were highly coincident with each other. Therefore, the model can be considered for use in simulating moving laser-polished alumina ceramics with a high degree of accuracy.

## 6. Conclusions

In this study, the optimal polishing process parameters were obtained by combining experiments and numerical simulations using orthogonal tests. A two-dimensional model coupled with heat transfer and laminar flow was established to simulate the evolution of the surface morphology of laser-polished materials under the action of a moving heat source, and the numerical model was verified experimentally. The following conclusions were obtained:(1)The laser polishing experiment was conducted on the surface of alumina ceramics using CO_2_ pulsed laser. The optimum parameters obtained through orthogonal tests were average power *P* = 70 W, scanning speed *v* = 320 mm/s, and laser pulse frequency *f* = 2 kHz. The roughness of the material surface decreased from Ra = 1.624 μm to Ra = 0.549 μm, down by 65.7%.(2)The evolution of the surface morphology of the polished melt pool was studied by establishing a two-dimensional numerical model. The simulation results showed that when the pulsed laser acted on the material surface, the fluid flowed from the peak to the trough under the action of capillary forces. Most of the area in the melt pool was dominated by capillary forces, and the fluid flow velocity was related to the undulation of the surface morphology.(3)Through comparison of the surface profile at the same position on the material surface, it was found that the surface bumps and depressions were gradually smoothened after polishing. The maximum depths of the numerical simulation and experimental polishing depressions at a distance of about 350 μm away from the starting point of polishing were −7.9 and −6.8 μm, respectively, with an error of 13.9%, while the maximum heights of the surface contour bumps at a distance of about 420 μm away from the starting point of polishing were 4.67 and 5.5 μm, respectively, with an error of 17.8%. The results show that the numerical model can be used to simulate laser polishing of alumina ceramics with high accuracy.

## Figures and Tables

**Figure 1 micromachines-14-02012-f001:**
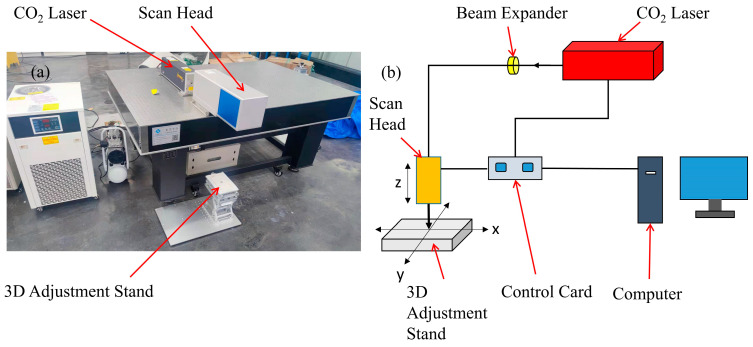
(**a**) Experimental setup, (**b**) schematic diagram of the experiment.

**Figure 2 micromachines-14-02012-f002:**
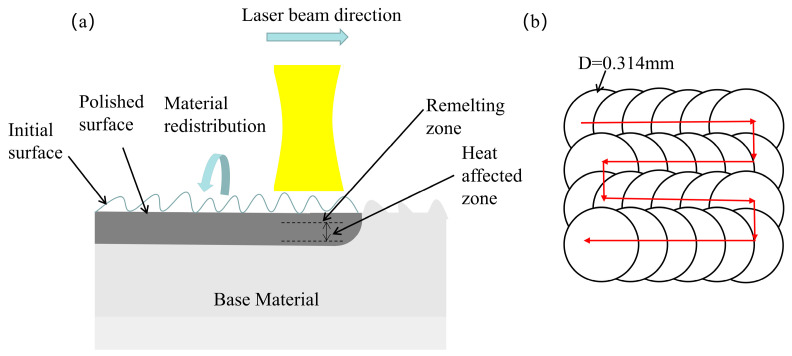
(**a**) Schematic diagram of laser polishing, (**b**) polishing pathway.

**Figure 3 micromachines-14-02012-f003:**
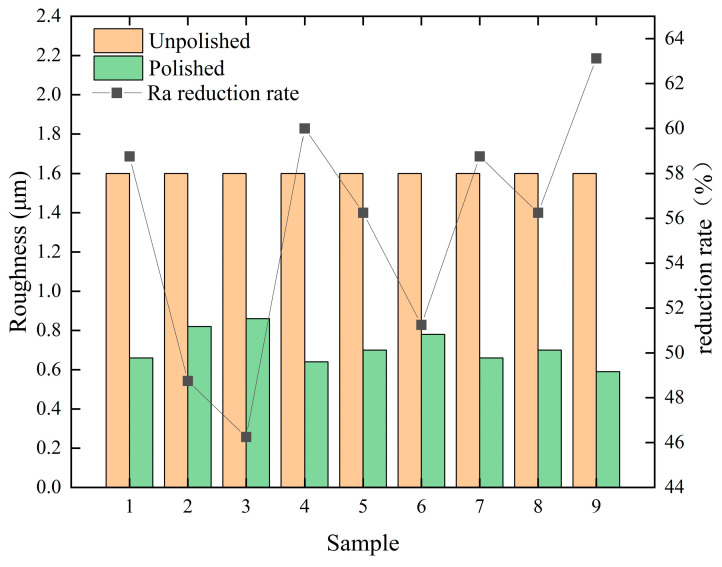
Variations of surface roughness in each experiment before and after laser polishing.

**Figure 4 micromachines-14-02012-f004:**
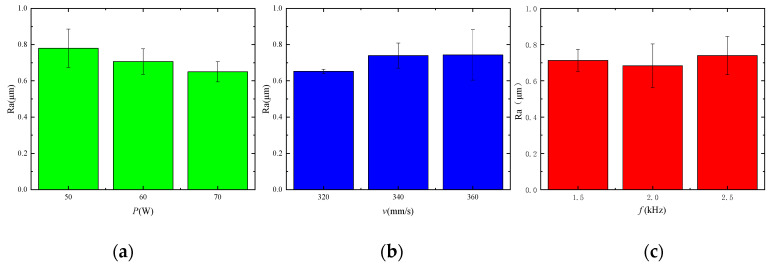
Range plot. (**a**) Range variation of average power, (**b**) range variation of the scanning speed, (**c**) range variation of the pulse frequency.

**Figure 5 micromachines-14-02012-f005:**
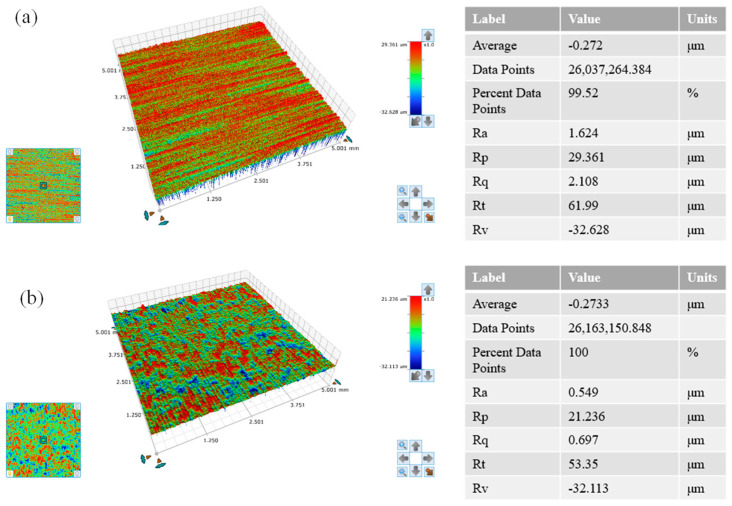
Three-dimensional morphology of the surface before and after polishing (**a**) Initial three-dimensional morphology of the material surface, (**b**) three-dimensional morphology of the material surface after polishing.

**Figure 6 micromachines-14-02012-f006:**
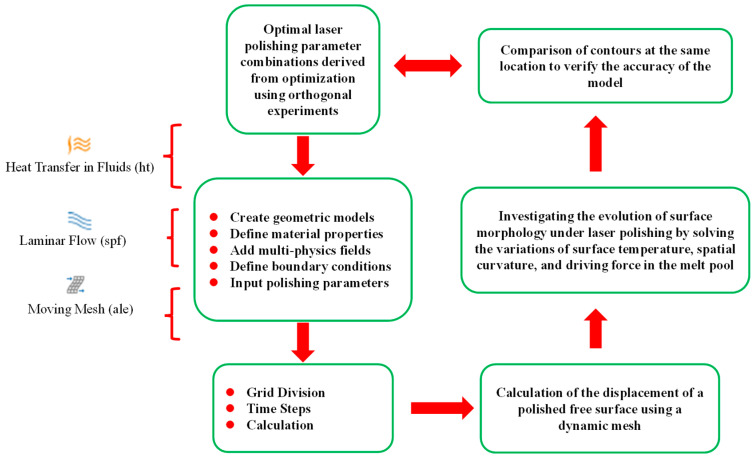
Numerical simulation flow chart.

**Figure 8 micromachines-14-02012-f008:**
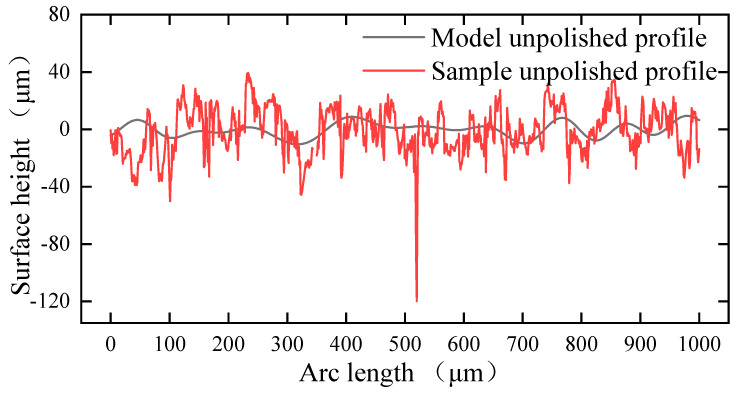
Surface profile before and after filtration.

**Figure 9 micromachines-14-02012-f009:**
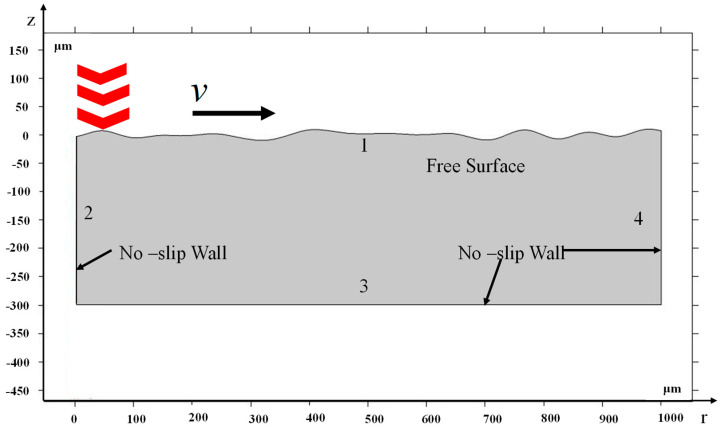
Geometric model.

**Figure 10 micromachines-14-02012-f010:**
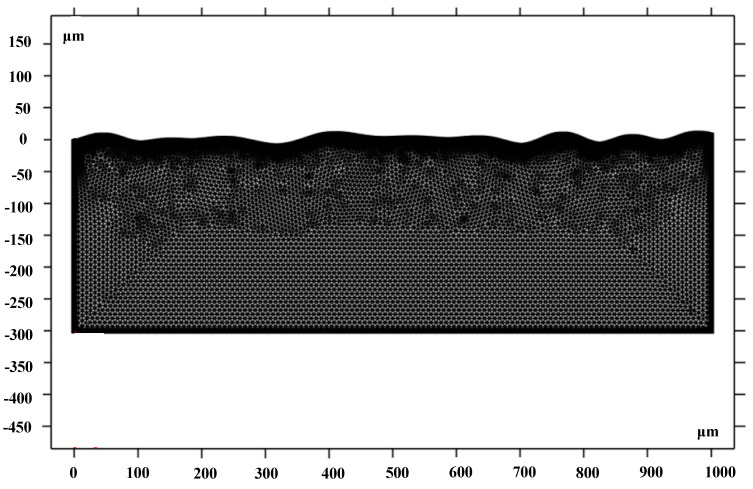
Geometric mesh.

**Figure 11 micromachines-14-02012-f011:**
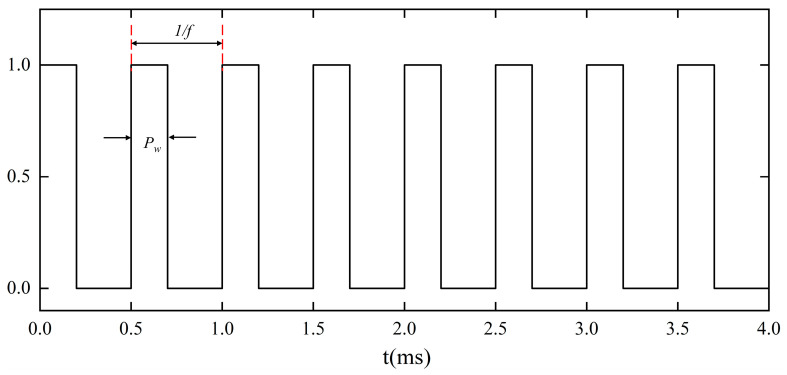
Square-wave pulse.

**Figure 12 micromachines-14-02012-f012:**
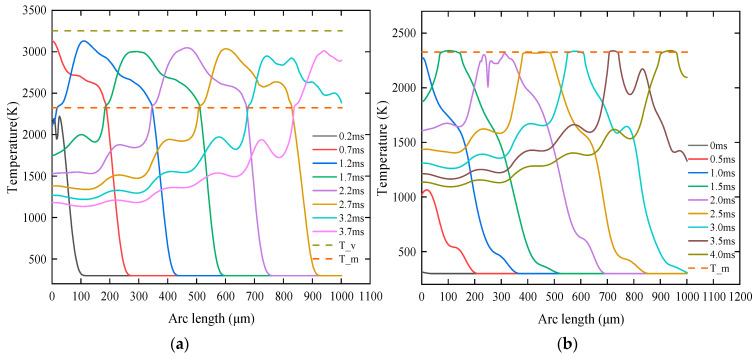
Distribution of material surface temperature at different times. (**a**) Distribution of material surface temperature at the pulse termination time, (**b**) material surface temperature at the pulse starting time.

**Figure 13 micromachines-14-02012-f013:**
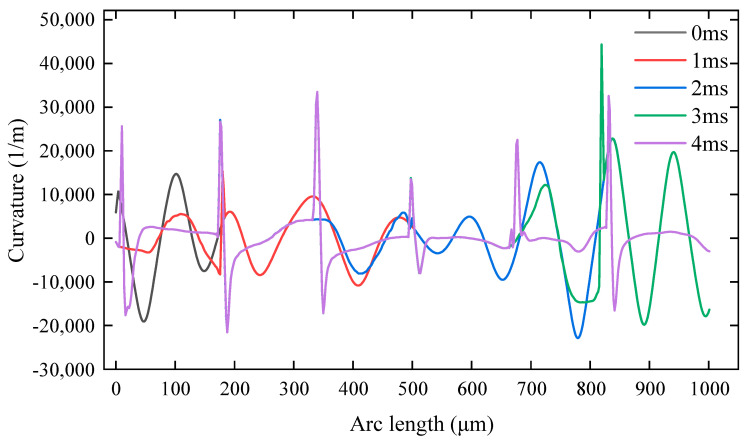
Spatial curvature.

**Figure 14 micromachines-14-02012-f014:**
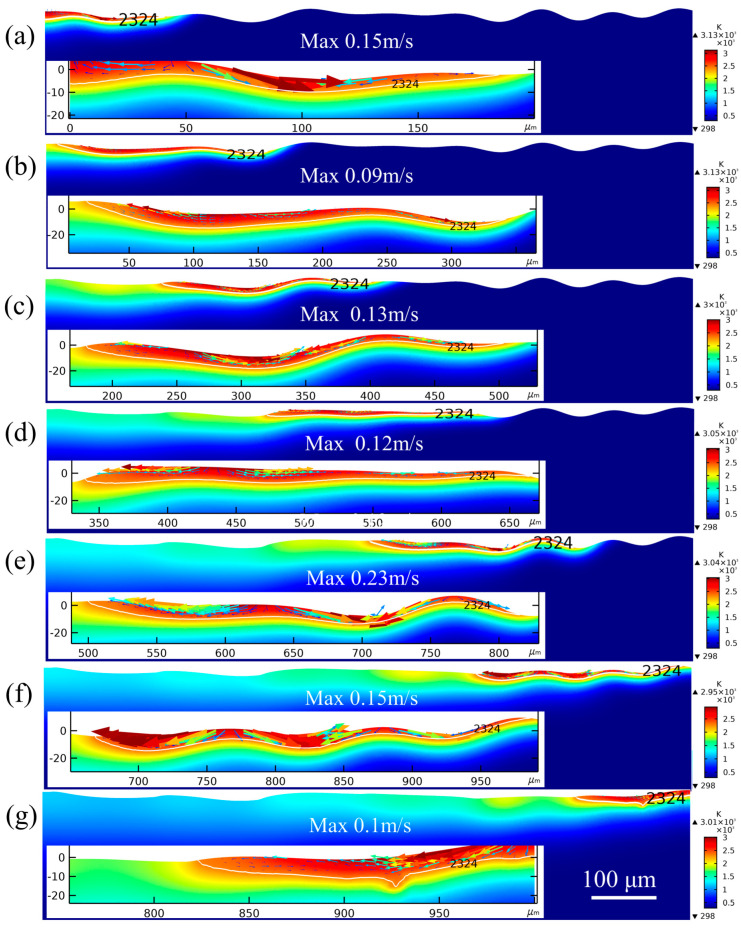
Evolution of the surface morphology of laser polishing material: (**a**) 0.7 ms, (**b**) 1.2 ms, (**c**) 1.7 ms, (**d**) 2.2 ms, (**e**) 2.7 ms, (**f**) 3.2 ms, (**g**) 3.7 ms.

**Figure 15 micromachines-14-02012-f015:**
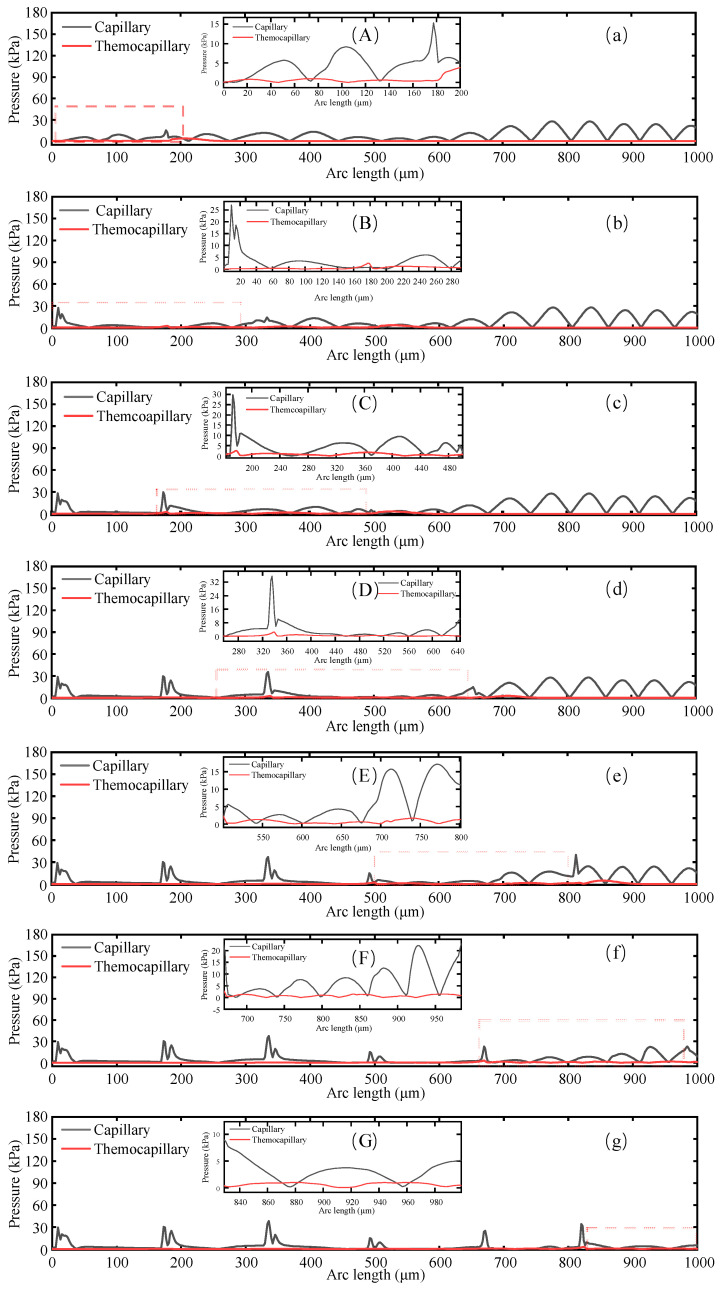
Surface distribution of capillary force and thermocapillary force in the melt pool at different times. (**a**) 0.7 ms, (**b**) 1.2 ms, (**c**) 1.7 ms, (**d**) 2.2 ms, (**e**) 2.7 ms, (**f**) 3.2 ms, (**g**) 3.7 ms. (**A**–**G**) shows the localized magnification of the corresponding moments.

**Figure 16 micromachines-14-02012-f016:**
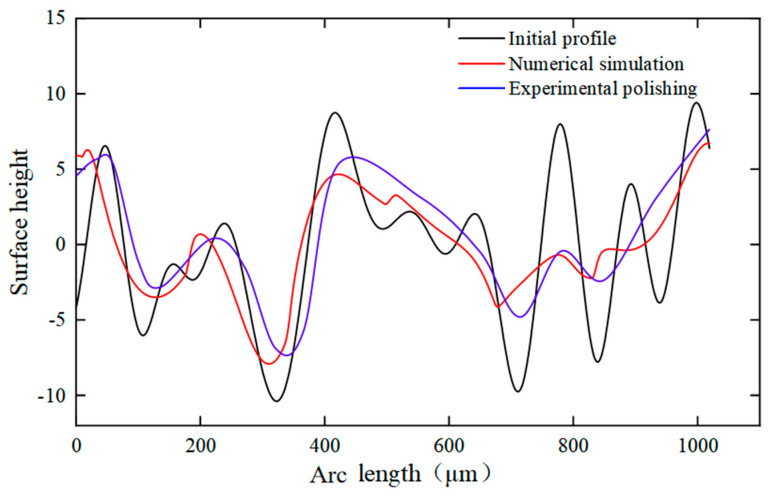
Variation of alumina ceramic surface profile under numerical simulation and experimental polishing.

**Table 1 micromachines-14-02012-t001:** Main chemical composition of 99 alumina ceramics.

Elements	Al_2_O_3_	Na_2_O	Fe_2_O_3_	SiO_2_	MgO	TiO_2_	CaO
Contents	99%	0.0776%	0.0124%	0.0238%	0.0521%	0.0035%	0.0136%

**Table 2 micromachines-14-02012-t002:** Factor levels for the orthogonal tests.

Variable	Code		
	Low	Medium	High
Laser power *P*/W	50	60	70
Laser scanning speed *v/*(mm∙s^−1^)	320	340	360
Laser pulse frequency *f*/kHz	1.5	2	2.5

**Table 3 micromachines-14-02012-t003:** Design matrix and result of the orthogonal test.

No.	*P*/W	*v*/(mm/s)	*f*/kHz	Ra_1_/μm	Ra_2_/μm	Ra_3_/μm	Ra/μm
1	50	320	1.5	0.73	0.67	0.60	0.66
2	50	340	2	0.79	0.82	0.84	0.82
3	50	360	2.5	0.80	0.94	0.85	0.86
4	60	320	2	0.70	0.58	0.63	0.64
5	60	340	2.5	0.64	0.74	0.70	0.70
6	60	360	1.5	0.73	0.74	0.86	0.78
7	70	320	2.5	0.72	0.67	0.60	0.66
8	70	340	1.5	0.68	0.69	0.74	0.70
9	70	360	2	0.58	0.57	0.61	0.59

**Table 4 micromachines-14-02012-t004:** Range analysis.

	Average Power *P*/W	Scanning Speed *v*/(mm/s)	Pulse Frequency *f*/kHz
K_1_	0.780	0.653	0.713
K_2_	0.707	0.740	0.683
K_3_	0.650	0.743	0.740
Range R	0.130	0.090	0.057

**Table 6 micromachines-14-02012-t006:** Boundary conditions.

Boundary Condition	Boundary No.	Physical Meaning	Equation	
Boundary heat source	1	Laser radiation	$I=\alpha \frac{Ppeak}{\pi {r_{0}}^{2}}\beta f\left(r_{1}\right)$	(9)
			$Ppeak=\frac{P}{Pwf}$	(10)
			$\beta =rect1(\mathrm{mod}(t,\frac{1}{f}))$	(11)
			$f\left(r_{1}\right)=\left\{\begin{array}{l} 0\;\left|r_{1}\right|\geq r_{0}\\ 1\;\left|r_{1}\right|\leq r_{0} \end{array}\right.$	(12)
			$r_{1}=r0+r-v\cdot t$	(13)
Convective heat flux	1, 2, 4	Free convection	$-k\nabla T=h(T_{a}-T)$	(14)
Surface-to-ambient radiation	1, 2, 4	Surface-ambient radiation	$-k\nabla T=\varepsilon \sigma (T_{a}^{4}-T^{4})$	(15)
Thermal insulation	3	Heat insulation	−k∇T=0	(16)
Weak contribution	1	Capillary force	$\sigma_{n}=\left(\nabla \cdot \vec{n}\right)\gamma \vec{n}=\kappa \gamma \vec{n}$	(17)
Marangoni effect	1	Thermocapillary force	$\sigma_{t}=\nabla \gamma =\frac{\partial \gamma }{\partial T}\nabla_{s}T\cdot \vec{t}$	(18)
Non-slip wall	2, 3, 4	Wall	dr=0,dz=0;dr=0	(19)

**Table 7 micromachines-14-02012-t007:** Parameters of the mesh size.

Parameter (Unit)	Top Layer	The Rest
Maximum element size (μm)	3	10
Minimum element size (μm)	0.05	0.1
Maximum element growth rate	1.05	1.2
Curvature factor	0.2	0.4

## Data Availability

The data presented in this study are available on request from the corresponding author.

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
