# Peer review of "Numerical Simulation and Validation of Laser Polishing of Alumina Ceramic Surface"

_micromachines, 2023, doi:10.3390/mi14112012_

Round 1

Reviewer 1 Report

Comments and Suggestions for Authors

Laser polishing has the potential  to change surface morphology considerably. Authors should focus on following points:

1. Have the authors used the commercially available CchO2 laser? If yes, what are the beam profile?

2. FIgure 7 needs improvement. Check x axis label.

3. Explanation in section 2.5 needs to be well connected with the figures. Please provide additional figures and correlate the contents.

4. If the roughness measurement has been done on surface area, better to denote it with Sa instead of Ra.

5. Section 3.3 has two different font sizes. Please maintain the homogeneity.

6. The simulation has been done with square beam while the experimental results belong to Gaussian beam. Please clarify.

Reviewer 2 Report

Comments and Suggestions for Authors

Dear Authors, the manuscript ‘Numerical simulation and validation of laser polishing alumina ceramic surface’, Manuscript ID: micromachines-2618657, has some weaknesses that must be revised appropriately.

Please find below some, of the most crucial comments:

1.      Some words in the “Abstract’ section indicating the meaning of the polishing or, generally, machining, in the industry applications should be placed.

2.      The main purpose and advantages of the study should be emphasized, staying with the ‘Abstract’ section.

3.      The sentence ‘Scholars from both home and abroad…’ is weird. Please rewrite it.

4.      The critical review in the ‘Introduction’ section does not exist. Only the advantages of the previous studies are presented. Especially according to the sentences ‘The aforementioned studies showed that there was an insufficiency of research focusing on the numerical simulation of laser polishing ceramic surfaces. By combining experiment and numerical simulation, this paper aims to obtain optimum polishing parameters through the orthogonal test and establish a two-dimensional model to simulate the evolution of the material surface morphology under a moving heat source. In order to verify the correctness of the numerical model, the size of the experimental melt pool is measured and compared with that of the simulated pool.’ this are not sufficient.

5.      The motivation for the work is not based on the lack of the current state of knowledge justified by the critical review. The Author(s) should highlight it against general words.

6.      More details should be provided in subsection 2.1, especially roughness measurements by the Bruker WLI instrument. How about the accuracy? No words against measurement uncertainty, noise or other errors. Please refer to those issues more comprehensively:

(1)   https://doi.org/10.1088/2051-672X/3/3/035004

(2)   https://doi.org/10.3390/coatings12060726

(3)   https://doi.org/10.1016/j.cirp.2014.03.086

7.      Some values in section 2.4 are not justified and look like selected arbitrarily. Please respond to this lack of reference appropriately.

8.      The flow chart of the numerical simulation experiment procedure should be added. Currently, it is difficult to retrieve what the Author(s) are trying to convey.

9.      The equations (1)-(8), if not newly proposed by the Author(s), should be referenced. In the presented form it is extremely difficult to separate what the Author(s) proposed and what is generally known. Further, the numbering of the equations on page 10 is false, where is doubled.

10.  Considering Figure 10, the Author(s) should explain the occurrence of an extraordinary values of the profile (deep individual peaks as valleys), like near the 500 µm length. Is this a micro-crack or some measurement error?

11.  The discussion, especially critical in section 4, is not appropriate. The author (s) should present some limitations of the study. In the current form, only the advantages are analysed. In that case, any further prospects cannot be even mentioned.

12.  Additionally, to the previous comment, section 5 is not sufficient and must comprehensively contribute to the advantages and disadvantages of the proposal. The Author(s) should emphasize the limitation of the proposed novelty.

13.  An additional, general conclusion must be provided for the ‘Conclusion’ section. The author (s) presented details information but the Reader(s) must receive one, main purpose and idea for the study.

14.  Additionally, the full DOI links should be added for all of the cited references, if exist.

Generally, the proposed manuscript can be classified as interesting, providing some crucial information on the topic addressed, however, includes some weaknesses that must be reduced appropriately.

Therefore, in its current form, the manuscript is not suitable for publication in a quality journal as the Micromachines is, requiring significant improvements.

Reviewer 3 Report

Comments and Suggestions for Authors

In this manuscript, the authors examine the influence of three laser parameters on the laser polishing effect, which are laser power, scanning speed and pulse frequency. At the same time, the authors establish a numerical model to investigate the temperature distribution and material flow in the laser polishing. It seems that these two parts are not closed related, as each of the part could serve as a stand-alone article. And what’s the purpose of building the numerical model since the information from the numerical model can be well predicted without the model?

1.       It should be “arc length” instead of “are length” in figure 11, 12, 14, 15?

2.       It is better to use energy density, i.e. power/area, to evaluate the laser power.

3.       Only two parameters, i.e. depth and height value of the profile at a specific location, are compared in the verification. More evidences are needed to verify the accuracy of the numerical model. And it should be a statistical value to evaluate a large area of the polished region.

4.       The numerical model should be used to reveal information or unknown patterns that can’t be observed experimentally. But here in the manuscript, the numerical model fails to deliver that goal.  

Comments on the Quality of English Language

The writing should be polished as several English errors are spotted in the manuscript.

Reviewer 4 Report

Comments and Suggestions for Authors

1.       The study investigates the effects of laser power, scanning speed, and pulse frequency on surface roughness. How did the authors determine the appropriate range and levels for these parameters since selecting an inadequate range or levels could lead to incomplete or inaccurate results?

2.       While orthogonal tests are useful for examining the effects of multiple factors simultaneously, they can be challenging to design and execute correctly. How did the authors consider the choice of the orthogonal array and the interaction between factors to ensure meaningful results?

3.       The authors use a numerical model established by COMSOL software to investigate material flow and surface morphology. Validating this model against experimental results is challenging, as there is a 17.8% maximum error between the numerical simulation and the experimental results, which is a lot. Achieving a higher level of accuracy in the model is essential.

4.       How are the authors sure about accurate modeling of the fluid flow in the melt pool during laser polishing which involves factors such as capillary forces and surface tension?

5.       The authors discuss obtaining "optimum polishing parameters" for reducing surface roughness. However, the generalizability of these parameters to different materials or conditions is not addressed. This limits the broader applicability of the findings.

6.       How did the authors manage to reproduce the experimental conditions and results when dealing with lasers and complex materials?

7.       The research relies on COMSOL software for numerical simulations. The accuracy and limitations of this software in simulating the specific laser polishing process are limited and need to be reconsidered.

8.       The study mentions that capillary forces dominate most of the area in the melt pool, but understanding the practical implications of this dominance and how it relates to surface roughness reduction is not discussed.

Comments on the Quality of English Language

As above.

Reviewer 5 Report

Comments and Suggestions for Authors

In this paper, the author established establish a two-dimensional numerical model of laser polishing alumina ceramic surface, which simulate the evolution of the material surface morphology under a moving heat source. It provides theoretical guidance for the study of laser polished ceramic surfaces However, the paper needs to be revised as follows before publication:

1. On page 4, the author noted that presents a comparison of the changed surface roughness in each experiment before and after laser polishing. It can be observed that the surface roughness is significantly reduced in the experiment 9, down by 63.125%. It is suggested that the author should explain in detail the reason the surface roughness is significantly reduced.

2. On page 5, the author noted that Table 4 shows that the R values are arranged from average power P, scanning speed v, and pulse frequency f., it is suggested that the author should mark error band in Figure 4.

3. On page 9, the author noted that boundary No.1,4 is surface-to-ambient radiation, and the boundary heat source is loaded on Boundary No.1. The author should explain what is the role of No 2? whether No.2 is surface-to-ambient radiation?

4. On page 13, the author noted that the curvature of the material surface profile is analyzed to reflect the polishing effect in Figure12. It is suggested that the author should explain why the surface curvature curve is intermittent at 0ms of laser polishing.

5. On page 17, the author noted that numerical simulation and experimental polishing surface contour of the maximum depth of the depression and convexity of the maximum height of the error of 13.9%, 17.8%, respectively. It is suggested that the author should explain and analyze the reasons for the error.

Comments on the Quality of English Language

check

Round 2

Reviewer 1 Report

Comments and Suggestions for Authors

Paper has been revised.

Reviewer 2 Report

Comments and Suggestions for Authors

The manuscript was improved substantially so can be accepted in the current, revised form.

Reviewer 4 Report

Comments and Suggestions for Authors

Accept in present form

Comments on the Quality of English Language

As above
